# Assessment of Diagnosis, Prognosis and Immune Infiltration Response to the Expression of the Ferroptosis-Related Molecule HAMP in Clear Cell Renal Cell Carcinoma

**DOI:** 10.3390/ijerph20020913

**Published:** 2023-01-04

**Authors:** Jing Leng, Zixuan Xing, Xiang Li, Xinyue Bao, Junzheya Zhu, Yunhan Zhao, Shaobo Wu, Jiao Yang

**Affiliations:** 1Department of Medical Oncology, the First Affiliated Hospital of Xi’an Jiaotong University, Xi’an 710061, China; 2Department of Infectious Diseases, The Second Affiliated Hospital of Xi’an Jiaotong University, Xi’an 710061, China

**Keywords:** HAMP, kidney cancer, biomarker, prognosis, immune infiltration

## Abstract

Background. Hepcidin antimicrobial peptide (HAMP) is a key factor in maintaining iron metabolism, which may induce ferroptosis when upregulated. However, its prognostic value and relation to immune infiltrating cells remains unclear. Methods. This study analyzed the expression levels of HAMP in the Oncomine, Timer and Ualcan databases, and examined its prognostic potential in KIRC with R programming. The Timer and GEPIA databases were used to estimate the correlations between HAMP and immune infiltration and the markers of immune cells. The intersection genes and the co-expression PPI network were constructed via STRING, R programming and GeneMANIA, and the hub genes were selected with Cytoscape. In addition, we analyzed the gene set enrichment and GO/KEGG pathways by GSEA. Results. Our study revealed higher HAMP expression levels in tumor tissues including KIRC, which were related to poor prognosis in terms of OS, DSS and PFI. The expression of HAMP was positively related to the immune infiltration level of macrophages, Tregs, etc., corresponding with the immune biomarkers. Based on the intersection genes, we constructed the PPI network and used the 10 top hub genes. Further, we performed a pathway enrichment analysis of the gene sets, including Huntington’s disease, the JAK-STAT signaling pathway, ammonium ion metabolic process, and so on. Conclusion. In summary, our study gave an insight into the potential prognosis of HAMP, which may act as a diagnostic biomarker and therapeutic target related to immune infiltration in KIRC.

## 1. Introduction

Clear cell renal cell carcinoma (ccRCC), also known as kidney renal clear cell carcinoma (KIRC), is among the most common malignancies worldwide, representing 80% of kidney malignancies [1]. As it is prone to metastasis, the prognosis of KIRC patients is poor, especially for patients in the late clinical stage [2,3]. One study showed that the 5-year overall survival rate (OS) of early stage KIRC could reach 96%, but it was no more than 10% for advanced stages [4]. Although advanced KIRC can be treated with molecular targeted therapy and immunotherapy, the long-term efficacy is still unsatisfactory [3]. Therefore, in order to prolong the overall survival of patients with KIRC there is a pressing need to screen potential diagnostic biomarkers, or identify the potential therapeutic targets related to tumorigenesis, metastasis and prognosis [5,6].

Notably, KIRC cells exhibit substantially higher sensitivity to ferroptosis than normal renal cells, which possess a basal level of ferroptosis sensitivity. The import, export, storage and turnover of iron impact ferroptosis sensitivity [7,8]. Moreover, previous studies have shown that the sensitivity of cells to ferroptosis is tightly controlled by iron homeostasis [9]. Nowadays, cancer cells are known to stockpile intracellular iron through dysregulation of the iron metabolism, partly by upregulating the genes involved in iron uptake [10]. HAMP, the hepcidin antimicrobial peptide gene, can produce hepcidin, which is a polypeptide hormone with an important role in the maintenance of iron homeostasis [11].

The HAMP gene is composed of three exons on chromosome 19q13.12 and yields the active hormone of hepcidin-25 peptide [12,13]. It is well known that HAMP plays an essential negative role in iron homeostasis, and its production of hepcidin was proven to be expressed in many cancers and non-cancer tissues [14,15]. In addition, a previous study showed that HAMP induced aggressive hepatocellular carcinoma (HCC), colorectal cancer (CRC) and skin cutaneous melanoma (SKCM), so we can infer that HAMP may be a potential prognostic and diagnostic biomarker for KIRC [16,17,18].

Moreover, hepcidin is reported to be an acute-phase protein involved in the innate immune reactions related to interleukin-6 (IL-6), conventional dendritic cells (cDCs) and so on [19,20]. Hepcidin can regulate the iron levels through the macrophages to influence inflammation, infection and possibly cancer [21,22]. In recent studies, iron metabolism within the tumor microenvironment has been uncovered to play a more critical role in tumors, especially in malignant transformation, and to affect both tumor-associated macrophages and tumor-infiltrating lymphocyte functions. Therefore, our study aimed to find how the expression of HAMP correlated to infiltrating immune cells and whether HAMP is the novel immune related therapeutic target for KIRC [23].

To confirm this inference, we used publicly available cancer databases and R programming to assess the expression of HAMP in KIRC and its prognostic and predictive role. Furthermore, we studied the link between HAMP expression and immune infiltration of tumors, and we evaluated the protein–protein network and genomic alterations by means of multi-dimensional analysis. Our results revealed the important function of HAMP in KIRC and confirms its possibility of becoming a potential biomarker for KIRC diagnosis and treatment.

## 2. Materials and Methods

### 2.1. Data Acquisition

We downloaded the gene expression profiles and clinical information from patients with renal clear cell carcinoma from the Cancer Genome Atlas (TCGA, https://tcga-data.nci.nih.gov/tcga/, accessed on 21 July 2022), a public repository of high-throughput experimental data. The type of data was RNAseq-FPKM, including 539 tumor tissues and 72 adjacent tumorous tissue samples.

### 2.2. Oncomine Database Analysis

The Oncomine database is one of the popular data-mining platforms, which we used to analyze the differential expression of HAMP in different cancer types [24]. In this study, the thresholds of the *p*-value, fold change and gene rank were, respectively, set as 0.001, 2 and the top 10%.

### 2.3. Timer Database Analysis

The Timer (Tumor Immune Estimation Resource) database is a comprehensive tool mainly used to analyze immune infiltrates by RNA-seq [25]. The website contains over 10,000 samples in 32 different cancer types from TCGA [26]. We studied the differential expression of HAMP between tumor and normal tissues through use of the “Diff Exp” module. In addition, we used the “Correlation” module to analyze the expression scatterplots between HAMP and the immune marker genes in KIRC. Moreover, the “SCNA” module was used to evaluate the differences in tumor infiltration with different copy number alterations for HAMP.

### 2.4. Ualcan

Ualcan is an online resource with an interface that allows the user to analyze the relative expression levels of a query protein or set of proteins across specific tumor sub-groups [27]. In the CPTAC confirmatory/discovery datasets, we analyzed the total protein expression levels of HAMP in KIRC.

### 2.5. GeneMANIA Database Analysis

GeneMANIA is a website for generating hypotheses about genes’ functions using the available genomic and proteomic data [28]. In our study, HAMP was submitted to the website to select the 50 most closely related genes.

### 2.6. STRING Databases Analysis

STRING is designed to analyze gene–gene and protein–protein interactions, which are shown in a PPI network [29]. We used the STRING database’s “multiple proteins” function to construct a PPI network of HAMP and its correlated genes.

### 2.7. GEPIA Database Analysis

GEPIA is a web server that provides expression analysis functions for TCGA and GTEx data to predict genes’ correlations in cancer [30]. To pick the closest neighboring genes, we inputted HAMP in KIRC to find and download the 100 most similar genes, which we later used to make an intersection with another group selected by R.

### 2.8. Cytoscape Software

Cytoscape is a software package which is often used to visualize and construct functional networks and make further explorations [31]. We inputted the file downloaded from STRING and formed a PPI, then used the “cytoHubba” app to select the top 10 hub genes.

### 2.9. Gene Set Enrichment Analyis

Gene set enrichment analysis (GSEA) is a computational method that determines whether a predefined set of genes has statistically significant, concordant differences between two biological states [32]. To evaluate whether the selected gene, HAMP, was expressed significantly differentially in KIRC, we divided the samples into the high and low groups according to the differences in the expression and selected the enriched pathways.

### 2.10. Statistical Analysis

The data in this study were mostly analyzed by R 3.6.3, a package downloaded from Bioconductor. We compared the expression of HAMP between tumors and normal tissues in paired and non-paired samples by Wilcoxon’s rank-sum test, and the difference showed statistical significance. The differences in the expression levels between Stages T1–T2 and Stages T3–T4 were evaluated through Wilcoxon’s rank-sum test. Various factors including sex, stage, age, and so on, were included to analyze the probability of survival by univariate and multivariate Cox regression. Results with *p* < 0.05 were considered to be significant.

To analyze the prognosis of KIRC, we used Kaplan–Meier curves to analyze the disease-specific survival, overall survival and progression-free survival under low and high expression levels of HAMP. Furthermore, we used the ROC and time-based ROC to evaluate the sensitivity of HAMP in diagnosis. The “Survival” and “survminer” packages in R were used in the survival analysis, the pROC and timeROC packages were used for the ROC analysis, and the ggplot2 package was used to visualize the results.

Next, we used the data from CIBERSORT to create a violin plot with the R package vioplot. We also compared the 22 types of immune cells with each other to make a heatmap to select the most correlated cells with the R package pheatmap. To analyze the Pearson correlations between genes and immune cells, the R programming ssGSEA was used to calculate the correlations between the expression levels of HAMP and tumor-infiltrating immune cells.

To find the correlated genes and construct a PPI network, we first applied a single-gene correlation screening, and created an intersection with the similar genes selected from GEPIA to find the superposition, which established a foundation for further study. Meanwhile, with the coincident gene group, we drew a co-expression heatmap with the R.ggplot 2 package.

## 3. Results

### 3.1. Differential Expression of HAMP in KIRC

Given that HAMP may be a potential biomarker for KIRC, we first used the Oncomine database to analyze the mRNA expression levels of HAMP in multiple types of cancer and normal tissues. Significantly, higher expression levels of HAMP were found in brain, breast, color, kidney and lung cancers compared with the adjacent normal tissues (Figure 1A). 

After removing the tumors without data for comparison, among all the TCGA tumors HAMP was upregulated in breast cancer (BRCA), cervical cancer (CESC), colon cancer (COAD), large B-cell lymphoma (DLBC), esophageal cancer (ESCA), glioblastoma (GBM), head and neck cancer (HNSC), kidney chromophobe (KICH), kidney clear cell carcinoma (KIRC), kidney papillary cell carcinoma (KIRP), low-grade glioma (LGG), lung adenocarcinoma (LUAD), lung cancer (LUSC), ovarian cancer (OV), rectal cancer (READ), skin cutaneous melanoma (SKCM), stomach cancer (STAD), testicular cancer (TGCT), thyroid cancer (THCA), thymoma (THYM), endometrioid cancer (UCEC) and uterine carcinosarcoma (UCS), but was downregulated in bile duct cancer (CHOL), liver cancer (LIHC) and pancreatic cancer (PRAD) compared with the corresponding normal tissues (Figure 1B).

Furthermore, as Figure 1C shows, the median expression level of HAMP in tumors is much higher than in normal tissue. To eliminate other mixing factors we analyzed paired samples, and the trend was consistent and maintained a significant difference (Figure 1D).

Since epigenetic alterations have been proven to play a role in cancer biology, we also explored the promoter methylation level of HAMP in KIRC and normal samples with UALCAN. Interestingly, the expression level in primary tumors was significantly lower compared with normal tissues, and the normal-vs.-primary statistic was 1.62 × 10^−12^ (Figure 1E).

### 3.2. The Clinical Correlation and Prognostic Value of HAMP in KIRC

To assess the correlation between the expression of HAMP and clinical pathologic outcomes in tumors, we explored the pathological stage of KIRC patients. The results revealed that the gene HAMP was significantly upregulated in Stages T3–T4 compared with Stages T1–T2 (Figure 2A).

In addition, we constructed a nomogram to visualize the prognostic model of the Cox regression analysis and found that the concordance was 0.762 (0.738–0.786) (Figure 2B). Further, we performed univariate and multivariate Cox regressions to find the correlations between HAMP and age, stage, etc. (Appendix A).

To evaluate the prognostic significance of HAMP in KIRC, we compared the patients with high and low expression levels in terms of overall survival (OS), disease-specific survival (DSS) and progression-free interval (PFI), and the results were HR = 1.53 (1.55–3.51; *p* < 0.001) for DSS, HR = 1.82 (1.33–2.48; *p* < 0.001) for OS and HR = 1.68 (1.22–2.32; *p* = 0.001) for PFI. The outcomes indicate that HAMP is a dangerous factor for KIRC, which plays an important role in prognosis (Figure 2C–E).

Furthermore, we used the ROC to construct a model to predict the probability of survival and found that the AUC was 0.911 (0.879–0.944) in the ROC curve, which showed that HAMP is a significant predictive index in diagnosis (Figure 2F). Similarly, the AUC in the ROC for OS was 0.649 (0.601–0.698) and the AUC of the time-dependent ROC predicted that the 1-year, 3-year and 5-year outcomes were, respectively, 0.686, 0.631 and 0.627 (Figure 2G,H).

### 3.3. Correlation between Immune Cells and HAMP Expression Levels in KIRC

As we know, tumor-infiltrating immune cells are one of the representative cellular components of the host’s anti-tumor immune responses and tumor immune escape [33]. Therefore, we explored the correlation between HAMP expression levels and immune infiltration levels to confirm the effect of HAMP in prognosis in KIRC. According to the HAMP expression levels, we divided the samples into the tumor and normal expression groups and analyzed the differences in 22 immune cells. The results revealed that CD8T cells, CD4 memory resting T cells and M2 macrophage cells were actively expressed, and regulatory T cells (Treg) and neutrophils were closely related to the HAMP expression levels. Interestingly, regulatory T cells (*p* = 0.012) were much higher in the normal tissue group than in the tumor group, while neutrophils (*p* = 0.015) showed a completely opposite pattern (Figure 3A). 

In addition, a correlation heatmap was used to visualize the relational degree within the subpopulation of immune cells. CD8 T cells were the most negatively related to CD4 memory resting T cells and closely positively related to regulatory T cells (Figure 3B).

We also used a lollipop chart to clearly compare the degree of correlation among the 24 immune cells and selected the most representative immune cells for further exploration. Moreover, we used the TIMER database to analyze the correlation between the expression levels of HAMP and the representative immune markers (Table 1). As shown in Figure 4A, the top 8 immune cells selected in Figure 3C with a Pearson correlation of >0.4 all had a linear relationship with the expression levels of HAMP (Figure 3C). 

To further find the mechanism of how HAMP expression acts on the immune cells, we drew scatterplots between HAMP expression levels and the immune marker genes of M1/M2 macrophages, T cells and B cells in KIRC based on the TIMER database. The results showed statistical significance (Figure 4B–E). Moreover, the correlations between HAMP and the related marker genes of B cells, T cells and macrophages in KIRC are listed in Table 2.

After we had analyzed the somatic copy number alterations (SCNAs) in KIRC, there were significant differences in B cells, CD8+ T cells, CD4+ cells, neutrophils and dendritic cells, which confirmed the prediction that HAMP may regulate the tumor process through immune infiltration (Figure 4G). We also assessed the correlations between the most common immune checkpoints and the expression levels of HAMP to further strengthen the potential mechanism between the expression levels of HAMP and immune cells (Figure 4F). 

### 3.4. The Network of HAMP Expression in KIRC

In cancer research, constructing a PPI network is a useful method for revealing co-expressed and related genes [34]. We constructed a PPI network of HAMP and another 50 interacting proteins in GeneMANIA. (Figure 5A) Moreover, we used the GEPIA database and R programming to find the top 100 similar genes of HAMP and constructed a Venn diagram to make an intersection (Figure 5C). There were 39 co-expressed genes, as shown in Appendix A. The interactions of the 39 genes and HAMP were visualized with STRING.

In addition, we plotted a heatmap of HAMP’s co-expressed genes to analyze the correlations, and the results showed a highly consistent trend, which means that these 39 co-expressed genes have significant positive correlations with the key gene, HAMP (Figure 5D).

We used the “cytoHubba” module on the PPI network to calculate the nodes’ scores and selected the top 10 hub nodes as ranked by MCC. The key genes extracted and the shortest paths between the hub genes are displayed in Figure 5E.

### 3.5. Enrichment Analysis of HAMP in KIRC

As GSEA has been extensively utilized in the context of differential expression analysis, we also analyzed the intersection genes described above to uncover the likely functional interpretations [35]. Gene Ontology (GO) is commonly used for describing our knowledge of the biological domain with respect to three aspects: molecular function, biological processes, and location in the cellular component [36]. In total, 269 GO terms (251 terms for BP, 10 terms for CC and 8 terms for MF) were enriched, with an adjusted *p*-value of <0.05 and a q-value of <0.2. For biological process, regulation of lymphocyte activation, synapse pruning, and phagocytosis were enriched. For the cellular component, tertiary granules, collagen trimers and secretory granule membranes were the main locations. In the MF category, amyloid-beta binding, immunoglobulin binding and IgG binding were enriched. In addition, KEGG enrichment analysis was applied to find some pathways where the genes were enriched, such as Fc gamma R-mediated phagocytosis, *Staphylococcus aureus* infection, osteoclast differentiation, and so on (Figure 6A). Based on the GO and KEGG pathway enrichment analyses, the networks of the genes, terms and their interactions were visualized (Figure 6B).

In addition, based on the groups with low and high expression levels of HAMP in KIRC, we explored the HAMP-related signaling pathways in the GO and KEGG enrichment analyses, which showed significant differences (FDR < 0.05 and *p* < 0.05). The selected pathways in the groups with high and low expression are listed in Appendix A. The KEGG pathway analysis revealed two pathways that were positively correlated with high expression levels of HAMP, namely the oxidative phosphorylation and T cell receptor signaling pathways, and six negatively related pathways: autoimmune thyroid disease, the B cell receptor signaling pathway, Huntington’s disease, the JAK-STAT signaling pathway, non-small cell lung cancer, pancreatic cancer and small cell lung cancer (Figure 6C). Two GO items, namely the ammonium ion metabolic process and beta catenin binding, showed an upregulation of HAMP according to the FDP q-values and NOM *p*-values (Figure 6D).

## 4. Discussion

HAMP is closely related to ferroptosis, which is a key factor of the iron overload disease hemochromatosis [37]. It is reported ubiquitously expressed in cancers and can, to some extent, be a cancer-driving gene [38]. We found that HAMP was highly expressed in KIRC, KIRP, LUAD, SKCM, and so on, which corresponds to Liu’s former study [13]. Significant differences between normal and tumor tissues in KIRC were explored. In a previous study, KIRC patients had a poor prognosis, especially those in advanced T stages [3]. Our study has proven the overexpression of HAMP in later stages of the disease. We also observed that the expression level of HAMP is closely related to the probability of survival and the higher the expression, the worse the DDS, OS and PFS. In addition, our results also showed the mechanism of how HAMP affects prognosis, namely the interactions among the expression level of HAMP and the degree of immune infiltration and different immune markers. Based on all these results, HAMP can be considered as a potential diagnostic and prognostic biomarker and could be an immune-related treatment target.

We first analyzed the mRNA expression levels of HAMP in different cancers with Oncomine and R programming based on the TCGA database. There were discrepancies in the differential expression levels of HAMP in different cancer types among the different databases, which may reflect the differences in the data gathering methods and the fundamental mechanisms in terms of biological characteristics. With the same results in different methods, we can safely draw the conclusion that HAMP plays a significant role in cancers. In addition, it can be said that methylation in the promoter genes related to tumors is significantly connected to the clinical behavior of cancers [39,40]. Our UALCAN results showed the lower promoter methylation level of HAMP in KIRC, further confirming HAMP as a dangerous factor for epigenetic alteration.

Moreover, since the mRNA and protein expression levels of HAMP varied between normal and tumor tissues in KIRC, it showed high prognostic potential. In terms of prognosis, the T stage is one of the well-known prognostic factors, and a higher T stage indicates poor prognosis [41]. Upregulated expression levels of HAMP are related to the clinical stage and are highly likely to be a key factor in a lower probability of survival. Our study also found that a high expression level of HAMP is related to the diagnosis of KIRC. However, further steps are needed to explore the specific mechanisms and find strong support.

As we know, the immune system plays a major role in the control and progression of cancer [37]. Some studies have also reported that immune infiltration has an influence on the therapeutic responsiveness and prognosis of KIRC patients [42]. Thus, we have provided insight into the relationship between the expression levels of HAMP and immune cell infiltration in KIRC. In this study, the results showed that the expression level of HAMP has a converse correlation with the infiltration level of regulatory T cells and neutrophil. Former studies have shown that the neutrophil recruitment/activation results in tumor cell death, while Treg cells are recruited into the tumor microenvironment to mediate immune suppression, which is the reason for their positive and negative relationship, respectively, with the expression level of HAMP [43,44]. The results also showed that the expression level of HAMP has a strong positive correlation with the extent of infiltration of macrophages, Treg, Th1 cells, T cells and B cells. Moreover, the relationship between the expression level of HAMP and various immune cells’ gene markers indicates that HAMP could regulate the tumor immune microenvironment. The gene markers of M1 macrophages (NOS2, IRF5 and PTGS2) had a weak relationship with HAMP expression, whereas those of M2 macrophages were strongly correlated with HAMP, which predicted the regulatory function of HAMP in TAM polarization. In addition, the infiltration of a large number of Treg cells into tumor tissues is often associated with poor prognosis [45]. The increased expression of HAMP was found to have a significant correlation with Treg cells’ gene markers in KIRC. These findings suggested that HAMP plays a role in the recruitment and functioning on the immune infiltrating cells in KIRC.

After we constructed a PPI network of HAMP’s co-expressed genes in STRING and found their intersections in R, the collected 48 genes showed a strong relationship with HAMP. In addition, the selected 10 hub genes revealed a high degree of interaction, thus indicating potential biomarkers of KIRC. Based on the GO and KEGG pathway analyses, we found that HAMP was significantly associated with several immune response pathways and cancer pathways. Our study showed that HAMP is related to the biological processes of regulating lymphocyte activation, synapse pruning and phagocytosis, which correspond to the tumor immune environment. Moreover, the GO and KEGG pathways revealed the differentially enriched gene sets between the groups with high and low expression levels. The JAK-STAT signaling pathway and the related pathways were associated with increased mortality rates in renal cancer [46].

However, there are some defects and imperfections in our experiment. Although we used multiple databases for a comprehensive analysis, there were differences in the algorithms among the databases which may have led to deviations in the results. Similarly, as the data came from TCGA, we could not exclude whether the patients’ survival and other indicators were interfered with by external factors such as drug use. In addition, the specific mechanisms underlying the role of genes in cancer diagnosis remain to be explored, as the relationship between immune infiltration markers and gene expression is not sufficient to support this hypothesis. Nevertheless, our experiment is the first to investigate the effect of a high expression level of HAMP in KIRC on the prognosis of tumor patients, and it provides a possible target for clinical treatment.

## 5. Conclusions

In summary, our study has provided an insight into the potential prognostic use of HAMP, and thus it may act as a diagnostic biomarker and therapeutic target related to immune infiltration in ccRCC.

We found through bioinformatic analyses that the ferroptosis-related gene HAMP is a key gene in the KIRC. The expression of HAMP significantly increased in KIRC, and the evaluated expression levels of HAMP played a potentially important role in the diagnosis and prognosis of KIRC. Additionally, HAMP expression is positively related to immune infiltration in KIRC. Moreover, we found 39 genes that are similar to HAMP and selected 10 hub genes, which function in several KEGG and GO pathways.

## Figures and Tables

**Figure 1 ijerph-20-00913-f001:**
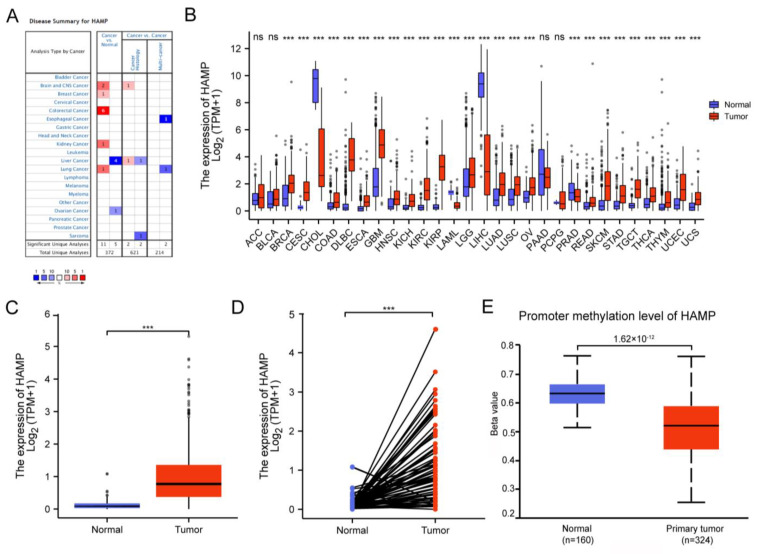
Differential HAMP expression levels in tumors and normal tissues. (**A**) Expression levels of HAMP in different cancers compared with normal tissues in the Oncomine database. (**B**) Comparison of the expression levels of HAMP in different cancers from the TCGA database. (**C**,**D**) The unpaired and paired expression levels of HAMP, compared between tumor ccRCC tissues (*n* = 539) and normal tissues (*n* = 79) in the TCGA database. (**E**) The hypermethylation level of HAMP in ccRCC tissues analyzed by the Ualcan database (*** *p* < 0.005).

**Figure 2 ijerph-20-00913-f002:**
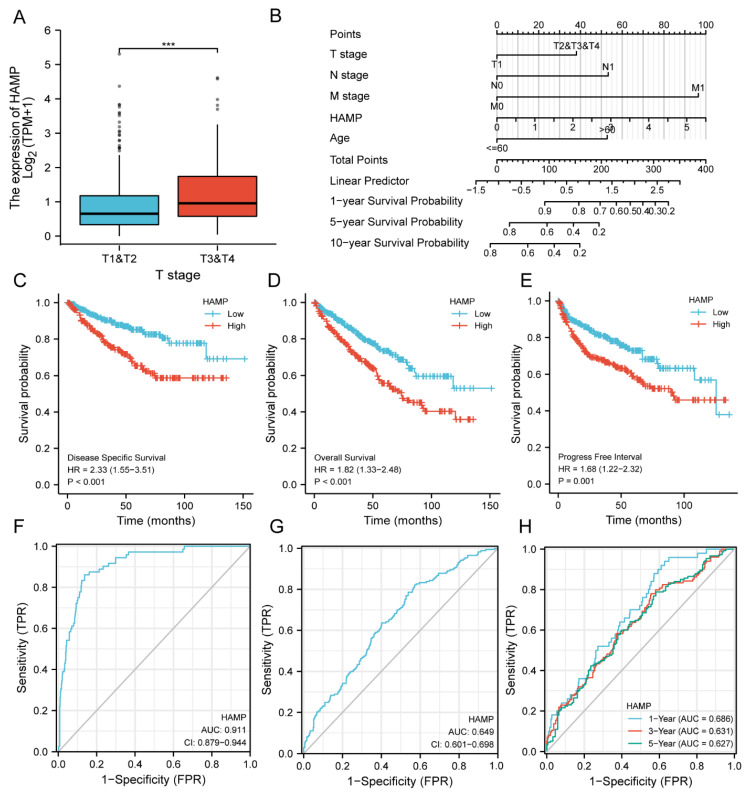
Prognosis and diagnosis based on HAMP expression levels. (**A**) The overexpression of HAMP in Stages T3−T4 compared with Stages T1−T2. (**B**) The scores of several factors with HAMP expression levels (T/N/M stage, age, 1/5/10 year survival probability). (**C**–**E**) Survival curves of OS, DSS and DFD correlated with HAMP expression levels in TCGA kidney cancer cohorts. (**F**–**H**) The diagnostic ROCs of status, OS and survival time based on HAMP expression levels in KIRC. OS, overall survival; DSS, disease-specific survival; PFI, progression-free interval. (*** *p* < 0.005).

**Figure 3 ijerph-20-00913-f003:**
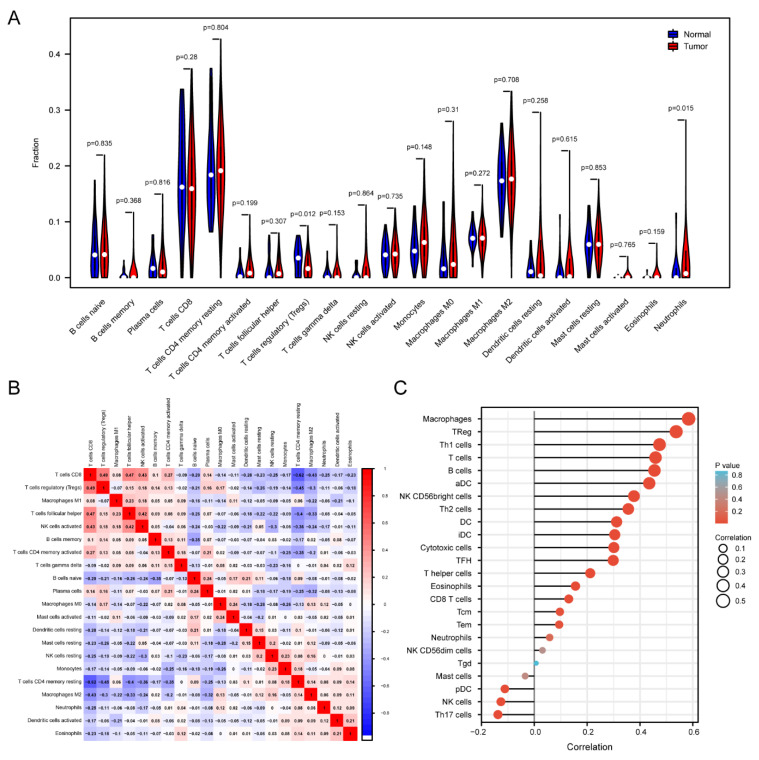
Infiltration of immune cells in KIRC patients. (**A**) The differences in 22 subtypes of immune cells between high and low expression levels of HAMP in kidney cancer. The group with a high expression level of HAMP is shown in red and the low−HAMP group is shown in blue. (**B**) Heatmap of the immune infiltration cells in tumor samples. (**C**) Lollipop chart comparing the degree of infiltration of the 24 immune cells in KIRC.

**Figure 4 ijerph-20-00913-f004:**
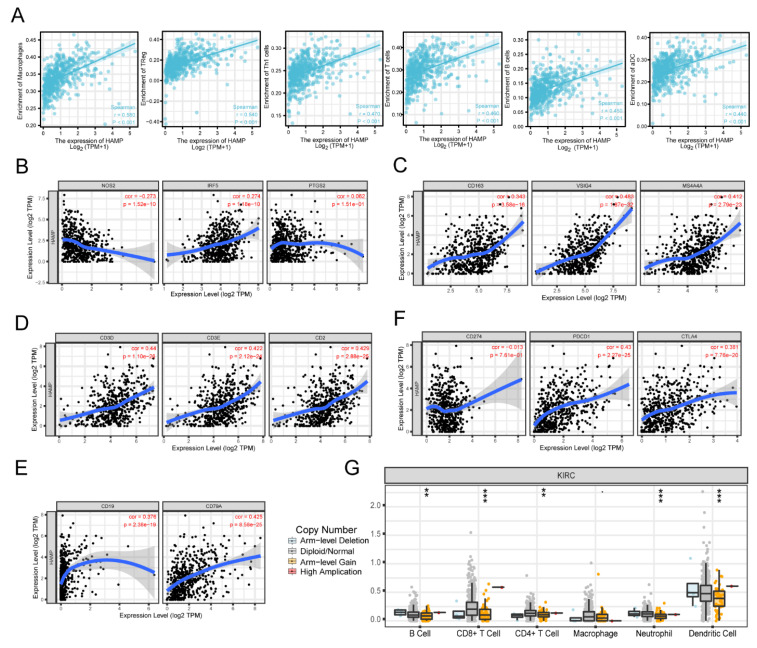
Correlation between expression levels of HAMP and immune infiltration levels in KIRC. (**A**) The enrichment in macrophages, Treg, Th1 cells, T cells, B cells and aDC based on the expression levels of HAMP in tumors. (**B**−**E**) Expression scatterplots between the expression levels of HAMP and the immune marker genes of M1/M2 macrophages, T cells and B cells in KIRC based on the TIMER database. (**F**) The immune checkpoint infiltration level versus the expression levels of HAMP in KIRC based on the TIMER database. (**G**) Significant differences in the somatic copy number alterations (SCNAs) in KIRC (** *p* < 0.01, *** *p* < 0.005).

**Figure 5 ijerph-20-00913-f005:**
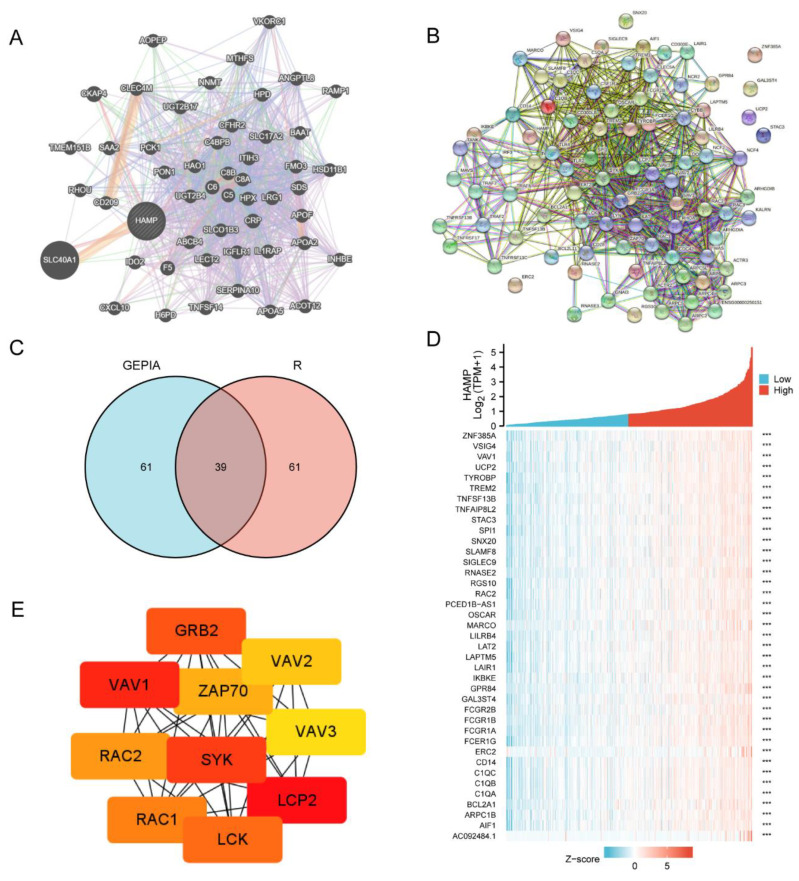
The PPI network and related genes of HAMP in KIRC. (**A**,**B**) The PPI networks of HAMP constructed by GeneMANIA and STRING. (**C**,**D**) The intersections of 39 co-expressed genes revealed by GEPIA and R programming, and their trends compared with the expression of HAMP. (**E**) Top 10 hub genes in HAMP expression selected by cytoHubba, based on the PPI and co-expression networks (*** *p* < 0.005).

**Figure 6 ijerph-20-00913-f006:**
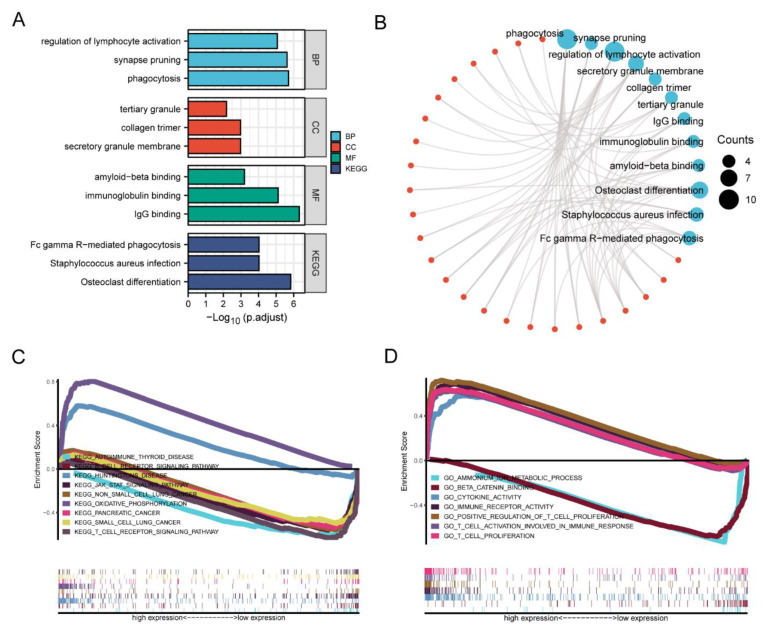
The functional enrichment of HAMP and its co-expressed genes in KIRC. (**A**) Bar plot of the top 3 GO enrichment terms for cellular components, biological processes, and molecular function, and KEGG enrichment terms. (**B**) The function network of GO and KEGG enrichment. (**C**,**D**) GSEA results, showing the differences in GO and KEGG enrichment between the groups with high and low expression levels in KIRC.

**Table 1 ijerph-20-00913-t001:** Correlation analysis for HAMP and the marker genes of immune cells in TIMER.

Cell Type	Gene Markers	KIRC			
		None		Purity	
		Cor	*P*	Cor	*P*
CD8+ T cell	CD8A	0.382	5.72 × 10^−20^	0.353	5.99 × 10^−15^
	CD8B	0.37	1.06 × 10^−18^	0.337	1.03 × 10^−13^
T cell (general)	CD3D	0.44	1.1 × 10^−26^	0.413	2.27 × 10^−20^
	CD3E	0.422	2.12 × 10^−24^	0.396	8.37 × 10^−19^
	CD2	0.429	2.88 × 10^−25^	0.402	2.33 × 10^−19^
B cell	CD19	0.376	2.38 × 10^−19^	0.337	1.02 × 10^−13^
	CD79A	0.425	8.58 × 10^−25^	0.396	1.13 × 10^−12^
Monocyte	CD86	0.546	0.17 × 10^−42^	0.537	7.34 × 10^−36^
	CD115 (CSF1R)	0.443	4.87 × 10^−27^	0.419	4.81 × 10^−21^
TAM	CCL2	0.016	7.18 × 10^−01^	−0.044	3.43 × 10^−01^
	CD68	0.456	0.01 × 10^−28^	0.477	1.57 × 10^−27^
	IL10	0.417	6.93 × 10^−24^	0.375	7.19 × 10^−17^
M1 Macrophage	INOS (NOS2)	−0.273	1.52 × 10^−10^	−0.332	2.43 × 10^−13^
	IRF5	0.274	1.18 × 10^−10^	0.273	2.64 × 10^−09^
	COX2 (PTGS2)	0.062	0.51 × 10^−01^	0.017	1.10 × 10^−01^
M2 Macrophage	CD163	0.343	3.58 × 10^−16^	0.34	6.06 × 10^−14^
	VSIG4	0.483	0.87 × 10^−32^	0.48	5.74 × 10^−28^
	MS4A4A	0.412	2.79 × 10^−23^	0.386	8.67 × 10^−18^
Neutrophils	CD66b (CEACAM8)	−0.107	1.34 × 10^−02^	−0.103	2.64 × 10^−02^
	CD11b (ITGAM)	0.434	7.38 × 10^−26^	0.436	9.54 × 10^−22^
	CCR7	0.366	2.32 × 10^−18^	0.342	4.51 × 10^−14^
Natural killer cell	KIR2DL1	−0.109	1.18 × 10^−02^	−0.114	1.44 × 10^−02^
	KIR2DL3	−0.074	8.94 × 10^−02^	−0.057	2.25 × 10^−01^
	KIR2DL4	0.177	3.86 × 10^−05^	0.154	8.88 × 10^−04^
	KIR3DL1	−0.135-	1.75 × 10^−03^	−0.115	1.34 × 10^−02^
	KIR3DL2	0.01	8.25 × 10^−01^	0.008	8.57 × 10^−01^
	KIR3DL3	0.026	5.5 × 10^−01^	0.011	8.16 × 10^−01^
	KIR2DS4	−0.096	2.73 × 10^−02^	−0.096	4.03 × 10^−02^

**Table 2 ijerph-20-00913-t002:** Correlation analysis between HAMP and the related marker genes of monocytes and macrophages in GEPIA.

Cell Type	Gene Markers	KIRC			
		Tumor		Normal	
		R	*P*	R	*P*
B cell	CD19	0.031	0.48	0.22	0.06
	CD79A	0.15	4.8 × 10^−4^	0.2	0.098
T cell	CD3D	0.24	2.9 × 10^−08^	0.62	5.2 × 10^−09^
	CD3E	0.26	2.2 × 10^−09^	0.55	4.2 × 10^−07^
	CD2	0.27	2.7 × 10^−10^	0.51	5.7 × 10^−06^
M1 Macrophage	INOS (NOS2)	−0.11	0.014	0.32	0.12
	IRF5	0.19	8.2 × 10^−06^	−0.037	0.76
	COX2 (PTGS2)	−0.021	0.64	0.082	0.49
M2 Macrophage	CD163	0.43	0	0.44	1 × 10^−04^
	VSIG4	0.59	0	0.49	1.3 × 10^−05^
	MS4A4A	0.37	0	0.48	1.9 × 10^−05^

## Data Availability

The data that support the findings of this study are openly available in The Cancer Genome Atlas (TCGA) at https://portal.gdc.cancer.gov, accessed on 21 July 2022. The datasets generated and/or analyzed during the current study are available in the Oncomine database (https://www.oncomine.org, accessed on 23 July 2022) the online tool Timer (https://cistrome.shinyapps.io/timer/, accessed on 23 July 2022), the online tool GEPIA (http://gepia.cancer-pku.cn/, accessed on 23 July 2022), the STRING website (https://cn.string-db.org/, accessed on 23 July 2022), the online tool Ualcan (http://ualcan.path.uab.edu/index.html, accessed on 23 July 2022) and the GeneMANIA website (http://genemania.org/, accessed on 23 July 2022).

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
