# Peer review of "Assessment of Diagnosis, Prognosis and Immune Infiltration Response to the Expression of the Ferroptosis-Related Molecule HAMP in Clear Cell Renal Cell Carcinoma"

_ijerph, 2023, doi:10.3390/ijerph20020913_

Round 1
Reviewer 1 Report
The author have mined available databases extensively to show that HAMP expression levels shows a co-relation to survival in KIRC patients and hence put forward the notion that HAMP can be used as a new prognostic bio-marker. They also draw correlations between HAMP and immune infiltrating cells which seems significant. I think this work is important if the following major issues can be addressed-
1. The legends for each figure need to be more detailed and lacks important information about the figures- The techniques used and the conclusions drawn.
2. The authors fail to explain why they have performed certain studies
a. Like why are they looking at promoter methylation in HAMP ? They themselves say that HAMP expressions varies amongst tissue types , so what information will promoter methylation give us ?
b.Why are they looking at Immune infiltration and HAMP ? Is there a significant study done to suggest that HAMP() iron metabolism) and Immune infiltration correlation is important? Like Neutrophils have a positive correlation to HAMP and rest mostly negative needs to be explained ? I think Fig 3 does not add value to the paper and should be removed if this point is not addressed.
3. The authors fail to cite an important reference -
doi - 10.3389/fonc.2020.01008.
This paper makes similar findings about HAMP and KIRC and needs to be cited. However the findings about HAMP expression levels and SKCM ( Fig 1 ) are opposites and it needs to be addressed why thats completely different.
4. Fig 1 - Other concerns.
Fig 1 B- not all cancers are paired as Tumour and Normal like for instance
ACC normal dataset is missing , Ov normal is also missing so thats needs to ev added.
Fig 1 E needs p values.
5. Grammatically errors are noted throughout the manuscript and needs to be corrected.
ex i Line 19 puts insights *
Author Response
Please see the attachment. Thank you !

Reviewer 2 Report
This manuscript describes the evaluation of diagnosis and prognosis through expression of the HAMP molecule in clear cell renal cell carcinoma. The authors describe the role of the HAMP molecule for the diagnosis of clear cell renal cell carcinoma, and it is very interesting data showing the potential as a new diagnostic biomarker. However, authors have to resolve some minor issues in order to be published in the IJERPH journal.
Minor revision
1. In general, abbreviations should appear in full name at the beginning of the paper. In general, abbreviations should appear in full name at the beginning of the paper. On line 25, authors must indicate the full name for KIRC.
2. In the introduction section, the authors do not provide a sufficient explanation of HAMP. It is recommended that an explanation of the association between HAMP and other immune cells in cancer be added.
3. In the Materials and Methods section, the authors did not describe the information and acquisition process of normal and tumor tissue.
4. In Figure 3A, the authors did not show p values for neutrophils. It is recommended to display the p value.
5. Authors need to edit the English language throughout the manuscript.
Reviewer 3 Report
Reviewer comments
Leng et al., studied potential prognosis using HAMP, a ferroptosis causing molecule that has differential expression in different types of cancer. Overall, this would be a potential promising biomarker to consider in developing targeted therapies, but there are some things that need to be addressed:
1. It would be better to list some abbreviations near the key words for better understanding.
2. Introduction can be improved further by adding clear connecting sentences.
For example: Page 2, line 46, the sentence “It is well known that the essential negative role of HAMP in iron homeostasis”, is there a relation to the next sentence?
3. There are numerous grammatical and punctuation mistakes throughout the manuscript that makes it harder to understand what is being said.
4. Figure1: Legends for normal and cancer colors are not indicative as per the figure. Some are light shades of blue, (C and D panels), some are purple (panel E) and then there are dark blue in panel B. Is that intentional? Clarify in the legends.
5. In Page 4, lines 150-151, the authors talk about different T stages which are not mentioned anywhere prior to this. Mention the specific reason as to why they were studied and analyzed.
6. Figure 3: Need to specify the different colors in the figures. I am assuming blue is normal and red is cancer. But needs clarification.
7. Figure 4G: The copy number colors are too small and harder to read in the legend beside the figure.
8. Figure 5A is very blurred and can’t really see.
9. As per the authors, how are the discrepancies using multiple different databases to assess the HAMP mRNA expression levels addressed? Clarify?
10. Double check and ensure that all the legends in the figures are properly addressed.
Author Response
Please see the attachment. Thank you !

Round 2
Reviewer 3 Report
The authors have answered and addressed all the comments.